# The Effect of the Weight and Type of Equipment on Shoulder and Back Muscle Activity in Surface Electromyography during the Overhead Press—Preliminary Report

**DOI:** 10.3390/s22249762

**Published:** 2022-12-13

**Authors:** Michalina Błażkiewicz, Anna Hadamus

**Affiliations:** 1Faculty of Rehabilitation, The Józef Piłsudski University of Physical Education in Warsaw, 00-809 Warsaw, Poland; 2Department of Rehabilitation, Faculty of Dental Medicine, Medical University of Warsaw, 02-091 Warsaw, Poland

**Keywords:** EMG, dumbbell overhead press, kettlebell overhead press, shoulder joint, resistance exercise, muscle action

## Abstract

The overhead press is a multi-joint exercise that has the potential to use a high external load due to the cooperation of many muscle groups. The purpose of this study was to compare the activity of shoulder and back muscles during the overhead press with a kettlebell and a dumbbell. Surface electromyography (EMG) for the anterior and posterior deltoid, upper and lower trapezius, serratus anterior, and spinal erectors was analysed for 20 subjects. Participants performed the four trials of pressing kettlebell and dumbbell, weighted at 6 kg, and 70% of one maximum repetition (1RM) in the sitting position. Statistical analysis was performed using a non-parametric Friedman test and a post-hoc test of Dunn Bonferroni. No significant differences were found in the activation of assessed muscles when comparing dumbbell to kettlebell press trials with the same load (6 kg and 70% of 1RM). However, muscle activity of all muscles except the upper trapezius was always higher for kettlebell pressing. Different center of gravity locations in the kettlebell versus the dumbbell can increase shoulder muscle activity during the overhead press. However, more studies are required to confirm these results.

## 1. Introduction

In strength sports, the primary motion that engages the muscles of the upper limbs are different types of pressing motion [1,2]. The two most popular pressing motions are bench and overhead. They differ not only in the alignment of body segments to each other but, above anything else, in the activity of the muscles involved in a given motion. The bench press is performed in the lying position, and the overhead press is in the standing or sitting position. This changes where the loading is relative to the spine—perpendicular for the bench press and inline for the overhead press. According to Stronska et al. [1], the bench press uses greater activation of the anterior deltoid and pectorals major muscles than the lateral head triceps brachii and long head triceps brachii at all exercise intensities (40% 1RM, 60% 1RM, 80% 1RM and 100% 1RM).

The overhead press is a complex movement that engages the muscles and articular structures of the shoulder girdle, upper arm, forearm and hand. Accordingly, this motion is very often used not only in strength training but also in rehabilitation. The initial phase of the overhead press involves flexion and internal rotation of the shoulder joint. It occurs due to the concentric work of the pectoralis major (clavicle part), coracobrachialis and anterior deltoid muscles. The internal rotation is constrained by the infraspinatus and teres minor muscles. Extension of the elbow joint provides concentric contraction of the triceps muscle of the arm with the simultaneous involvement of the ulnar muscle—an active stabilizer of the elbow joint, whose role increases when lifting loads [3]. When the flexion movement exceeds 60°, the serratus anterior, lower and upper trapezius muscles move the scapula externally [4,5,6]. The whole articular complex of the shoulder girdle is active from about 80–90° of shoulder flexion, including up to 30° of rotation in clavicle joints. After exceeding 90° flexion, trapezius, pectoralis minor, and rhomboid muscles are activated to provide necessary stabilization of the scapula. When the movement exceeds about 120°, the most engaged muscles are the serratus anterior and lower trapezius, which provide rotational movement of the scapula to support the overhead lift of the upper arm. At the same time, abdominal and back muscles are active to provide necessary core stabilization. The overhead press movement finishes with an extension and sideways flexion of the thoracic spine [3].

Many authors have examined the activation of the upper limb muscles during the overhead press [3,6,7,8,9,10,11,12,13,14]. This activation was presented according to the position of the body during pressing (sitting or standing) [10], surface stability (exercise bench or Swiss ball) [11], grip (narrow or wide) [12,13,14] and movement direction (behind or forward from the head) [3,14]. In addition, it is possible to find papers in which the authors analyze muscle activity depending on the type of equipment pressed: barbell vs. dumbbells [10,11], dumbbells vs. kettlebells [8], as well as the influence of exercise intensity on muscle activation [7,9].

The above-mentioned factors can affect the change of overhead pressing technique. Different techniques can affect the sequence of muscle activation and the magnitude of stress forces in ligaments and tendons. This can impact physiological adaptation changes forced by exercise [15]. The overhead press with the highest loads is performed with both hands using a barbell [15]. One-hand overhead presses are usually performed with kettlebells or dumbbells. One of the main features distinguishing kettlebells from dumbbells is the different locations of the center of mass. The center of mass of the dumbbell is within the handle. In contrast, the center of mass of a kettlebell is below the handle, within the ball. As a result, when pressing overhead, the center mass of the dumbbell is in line with the elbow joint. In contrast, when pressing a kettlebell, its center of mass is behind the joint [8]. This forces external rotation in the shoulder when the kettlebell is held at the side during an overhead press. Thus, it can increase the activation of the muscles that stabilize the shoulder [8,9]. The kettlebell is mainly used for ballistic movements such as swing, snatch and Turkish get-up [16]. However, dumbbells give more freedom to shoulder movements and therefore force higher stabilization of the shoulder than exercises with a barbell [3].

Ichihashi et al. [4] analyzed the kinematics of the scapula and clavicle during the military press exercise (one-hand dumbbell overhead press, main motion in frontal plane). They showed that movements of the scapula and clavicle during the military press differ significantly from those during shoulder flexion with and without weights. The kinematic features of the military press, which involved less scapular internal rotation, and higher upward rotation and posterior tilt than that during shoulder flexion, may make it useful as a re-education exercise for patients with scapular dyskinesia. Andersen et al. [9] examined scapular muscle activity during, among others, dumbbell overhead pressing performed at low and high intensities (Borg CR10 levels 3 and 8). They proved that only press-ups activated the lower trapezius and serratus anterior more strongly than the upper trapezius. These findings have important practical implications for exercise performance and preventing injuries in athletes. Paoli et al. [7] noted increasing muscle activation of the anterior deltoid, posterior deltoid and upper trapezius with increasing load on the dumbbells up to 70% of one-time maximum repetition (1RM). Saeterbakken and Fimland [10] showed higher activation of anterior and posterior deltoids during overhead press with a dumbbell compared to a barbell. However, Kohler, et al. [11] did not observe significant differences in deltoid muscle activation during the dumbbell and barbell overhead press. Activation of stabilizing trunk muscles has been assessed by two authors so far. Williams et al. [17] showed that the barbell overhead press with kettlebells suspended by elastic bands increased the activation of the erector spinae muscles when compared to the exercise with a static load. Kohle et al. [11] compared the activity of trunk stabilizing muscles during barbell and dumbbells overhead presses while sitting on a bench and Swiss ball. The erector spinae muscle showed the highest activity in pressing the barbell while sitting on a ball.

So far, only one paper considering muscle activity in the overhead press with a kettlebell compared to a dumbbell was published [8]. Dicus et al. [8] showed that overhead press in the horizontal plane (with a shoulder abducted to 90°) with dumbbell results with a higher anterior deltoid muscle activation than similar pressing with a kettlebell. Before now, no studies apart from Andersen et al. [9] have been published that answer the question of whether there are differences in muscle activation during overhead pressing in the sagittal plane (elbow joint facing forward) using kettlebells and dumbbells. Therefore, this study aimed to compare the activity of shoulder and back muscles, measured by surface electromyography, during the overhead press in the sagittal plane in a sitting position with a kettlebell and a dumbbell.

## 2. Materials and Methods

### 2.1. Participants

A group of 20 young men performing strength training at the amateur level with a minimum of 12 months of training experience participated in the study (Table 1). Only men were recruited for the study, because muscle activity patterns differ between women and men during the bench press depending on the external load [18]. Therefore, it can be assumed that a similar relationship will occur during overhead pressing. Participants were recruited from gyms where one of the authors (MB) is a fitness instructor. These included individuals preparing to complete Kettlebell instructor training or enthusiasts of this type of training. All participants were right-handed, clinically healthy, with full range of motion in the shoulder and elbow as well as without any reported history of upper-limb and lower-back muscle injury and neurological or cardiovascular disease in the previous 12 months. Range of motion in the shoulder and elbow joints was assessed using a goniometer in all anatomical planes [19,20].

The participants were asked to avoid caffeine and energy drinks in the 24 h preceding the study and 1RM measurement procedures. After a full explanation of the aims of the study and the experimental procedures, the participants signed written informed consent. They also had the opportunity to withdraw at any time. The project was approved by the Ethics Committee of the Józef Piłsudski Academy of Physical Education in Warsaw (SEK 01-09/2020) and conducted in accordance with the Declaration of Helsinki for research involving human subjects.

### 2.2. 1RM Measurement Protocol

For each participant, the procedure for determining 1RM took place seven and four days before the study for kettlebell and dumbbell pressing, respectively. During this time, the subjects did not perform any upper limb efforts or exercises. The 1RM assessment was performed using the same exercise technique that was later used in the study protocol.

Each participant sat on a box in the starting position without back support (Figure 1A). During all motion, the subject was required to maintain normal curvature in the lumbar, thoracic and cervical regions of the spine. The hip and knee joints were flexed approximately to 90°. All participants performed the overhead press by right-dominant hand. The left hand was on the left thigh all the time. The elbow joint of the right arm was in full flexion above the thigh and the wrist joint was in a neutral position. During the movement, the subject lifted the arm holding the elbow joint over the thigh at all times so that the main motion was in the sagittal plane. The elbow was straightened in the sagittal plane until the complete extension was achieved (Figure 1B) during both the kettlebell and dumbbell press.

The testing protocol started with a standardized warm-up, consisting of 3 sets × 10 repetitions with a load equal to 50% of the self-declared 1RM. This protocol was based on Brzycki [21] formula: *predicted 1RM = weight lifted / (1.0278 − 0.0278·n)*; where: *n* is the number of executed repetitions. Next, an additional load of about 5% more than the previous load was added until the subjects were unable to lift the dumbbell/kettlebell. Two minutes of passive recovery separated each trial. During the trial, each participant received standard encouragement from the operators.

### 2.3. Protocol

Before the measurements, the subjects performed a standardized warm-up protocol lasting about 5 min. The warm-up consisted of nine mobilization exercises for the joints of the upper limb, spine and lower limb girdle (Figure 2). Each of the nine exercises was repeated about 15 times. After warming up, subjects performed 10 repetitions of overhead presses with a dumbbell or kettlebell weighing 6 kg (Figure 1). It was necessary to verify the correctness of the technique.

The four test trials involved pressing overhead: (1) a 6 kg kettlebell (K-6kg); (2) a 70% of the 1RM kettlebell (K-70% 1RM); (3) a 6 kg dumbbell (D-6kg), and (4) a 70% of the 1RM dumbbell (D-70% 1RM). The 6 kg load was selected as a safe load that all participants could achieve. In people experienced in strength training, a higher load is advisable to activate highly susceptible motor units, so the subsequent load was 70% of the 1RM [22]. Each participant performed the trials twice in random order with the volitional cadence. The motion cadence was not specified because it changed the execution of the exercise to artificial (robotic). However, the timing of motion was assessed with a watch and averaged 3 s/0/2 s for all individuals. Trials with better technique (assessed by an experienced trainer) and without possible random errors (electrode falling off, cable entanglement) were taken for further analysis. Each individual rested for about 5 min between trials.

### 2.4. Muscle Activity Maesurement

During these trials, the activity of six muscles of the right upper limb (dominant) was examined using surface electromyography and parallel-bar EMG sensors. Surface electrodes (Ambu Blue Sensor N-00-S/25; Ambu A/S, Ballerup, Denmark) were placed following the SENIAM guidelines, with fixed 20 mm inter-electrode spacing on the following muscles: anterior deltoid, posterior deltoid, upper trapezius, lower trapezius, serratus anterior and spinal erectors—thoracis part. The skin around the designated areas was depilated, cleaned with alcohol band and dried with a sterile swab. The EMG signal was acquired at a sampling frequency of 1000 Hz using TeleMyo2400R G2 (Noraxon, Scottsdale, AZ, USA).

### 2.5. Data Analysis

The EMG signal was first band pass filtered (20–500 Hz, zero-phase 4th order Butterworth) and then processed using a root-mean-square algorithm. The average EMG envelope over a time window (RMS filter) was calculated with a 50 ms window size. A maximum value was selected for each trial (D-6kg, K-6kg, D-70% 1RM, K-70% 1RM) and each muscle activity. For statistical analysis, for each individual and each muscle, these values were normalized to the maximum value from among these four previously found [23]. The MVC test was not performed because maximum efforts were made during the measurements. EMG signal processing was done in Matlab software v. R2018b (MathWorks, Natick, MA, USA).

Statistical analysis was performed using PQStat 2021 software v. 1.8.2.238 (PQStat Software, Poznań, Poland). The normality of distribution was tested using the Shapiro-Wilk test and showed distributions different than normal in most cases. Therefore, a non-parametric Friedman test and post-hoc test of Dunn Bonferroni were used to find statistically significant differences between trials for specific muscles. The level of significance was set at *p* ≤ 0.05. The effect size was estimated using the Kendall’s W test value [24]. Kendalls uses the Cohen’s interpretation guidelines [25] to refer to effect sizes as small (0.1 < W< 0.3), moderate (0.3 < W< 0.5), and large (W > 0.5).

## 3. Results

Statistically significant differences were found between trials within the following muscles:(1)Anterior deltoid muscle: F (3, N = 80) = 23.71; *p* = 0.0001; W = 0.5647 (large).(2)Upper trapezius muscle: F (3, N = 80) = 13.65; *p* = 0.0034; W = 0.3250 (moderate).(3)Serratus anterior muscle: F (3, N = 80) = 20.23; *p* = 0.0001; W = 0.4817 (moderate).(4)Lower trapezius muscle: F (3, N = 80) = 20.14; *p* = 0.0001; W = 0.4795 (moderate).(5)Posterior deltoid muscle: F (3, N = 80) = 25.37; *p* = 0.0001; W = 0.6040 (large).(6)Spinal erector muscle—thoracis part: F (3, N = 80) = 30.39; *p* = 0.0001; W = 0.7236 (large).

After applying the post-hoc test, the results for each muscle are described in the following subsections.

### 3.1. Anterior and Posterior Deltoid Muscles

Statistically significant differences were found for both anterior and posterior parts of the deltoid muscle between the same trials. In both cases, medians of the maximum muscle activities were the highest for the K-70% 1RM and D-70% 1RM trials (Table 2). However, the exercise with a kettlebell with a load of 70% 1RM induced non-significantly higher anterior and posterior deltoid muscle activity than that recorded when pressing a dumbbell of the same weight. As for the anterior deltoid muscle, its activity increased by 19.04% concerning the activity recorded for the same weight dumbbell press, whereas the activity of the posterior deltoid muscle increased by 7.52%.

For 6 kg loads, the activity of anterior deltoid and posterior deltoid muscle increased by 0.88% and 3.63% for kettlebell work, respectively.

It is worth noting that the activities recorded during the K-70% 1RM trial were 76.99% and 81.81% higher than those recorded for the D-6kg trial for anterior deltoid and posterior deltoid muscles, respectively. Moreover, the activities of both muscles recorded during the K-70% 1RM trial was 75.43% higher than those recorded for the K-6kg trial.

In contrast, the activities recorded during the D-70% 1RM trial were 48.67% and 69.09% higher than those recorded for the D-6kg trial and 47.36% and 63.15% higher than those recorded for the K-6kg trial for anterior deltoid and posterior deltoid muscles, respectively. An example waveform of normalized muscle activity for anterior deltoid and posterior deltoid muscles is shown in Figure 3.

### 3.2. Lower and Upper Trapezius Muscles

The implementation of Dunn Bonferroni’s post-hoc test showed that the medians of maximum activities of the descending part of the trapezius muscle recorded for pressing a dumbbell and kettlebell of 70% 1RM were significantly higher by 10.28% and 12% than during the six-kilogram kettlebell press trial, respectively (Table 3).

For the ascending part of the trapezius muscle (Table 3), the values recorded for the 70% 1RM kettlebell press were significantly higher, by 55.03% and 40.84%, respectively, than those noted for the 6 kg dumbbell and kettlebell press attempts.

It is noteworthy that only for lower trapezius muscle, kettlebell exercise with both 6 kg and 70% 1RM loads induced non-significantly higher activity as compared to that reported during the same dumbbell weight press. Lifting a 6 kg kettlebell increased its activity by 10.07% and lifting a 70% 1RM kettlebell increased its activity by 24.22% in relation to the activity recorded for the same weight dumbbell press. The opposite was found for the median of maximum activation of upper trapezius muscle (Table 3). Non-significantly higher value was recorded for exercise with a dumbbell weighing 6 kilos. An example waveform of normalized muscle activity for upper and lower trapezius muscles is shown in Figure 4.

### 3.3. Serratus Anterior and Thoracis Part of Spinal Erector Muscle

The highest median of maximum activity of the serratus anterior muscles and spinal erectors was for the 70% 1RM kettlebell bench press test (Table 4). In this trial, the median of the maximum activity of these muscles was 72.41% and 24.22% higher, respectively, than when pressing a 6 kg dumbbell. Moreover, the median activation of these muscles was 45.98% and 22.69% higher when pressing a 6 kg kettlebell.

It is noteworthy that for both muscles, kettlebell exercise with 70% 1RM and 6 kg loads induced non-significantly higher median activity as compared to that reported during the same dumbbell weight press.

For the serratus anterior muscle, lifting a 70% 1RM kettlebell increased its activity by almost 30% concerning the activity recorded for a 70% 1RM dumbbell press. For the spinal erector muscle, lifting a 70% 1RM kettlebell increased its activity by 4.7% concerning the activity recorded for the weight dumbbell press. For the serratus anterior muscle, lifting a 6 kg kettlebell increased its activity by 18.1% concerning the activity recorded for dumbbell presses of the same weight. For the spinal erector muscle, lifting a 6 kg kettlebell increased its activity by 1.2% concerning the activity recorded for the same weight dumbbell press. An example waveform of normalized muscle activity for serratus anterior and spinal erector muscles is in Figure 5.

## 4. Discussion

The overhead shoulder press is an exercise involving multiple joints and muscles with the potential to reach high loads. This exercise is used in many sports and functional and rehabilitation training programs for the arm, shoulder, and scapula [6,7,26,27]. Moreover, it can be observed that, recently, exercises with kettlebells are replacing those with dumbbells. This study aimed to compare the activity of shoulder and back muscles during the overhead press with a kettlebell and the dumbbell with motion performed in the sagittal plane.

It is worth noting that, to date, most authors have analyzed the bench press mainly in a position where the upper limb was in abduction [7,8]. The presented work is one of the few [9] that analyzes this motion on positioning the upper limb in adduction. In this paper, the participants performed a kettlebell and dumbbell press with a 6 kg weight as a trial that was not very challenging. In addition, they performed the same movement with a load of 70% of one repetition maximum (1RM). The activity of six muscles: anterior and posterior deltoid, upper and lower trapezius, serratus anterior and spinal erector-thoracic part was examined. The reported large effect size for activation of the anterior and posterior deltoid muscles and spinal erector muscles, respectively, and moderate for the others, means that the differences found when pressing dumbbells and kettlebells with different weights for these muscles increase confidence in the findings.

No significant differences were found in the activation of assessed muscles when comparing dumbbell to kettlebell press trials with the same load (6 kg and 70% of 1RM). It was shown that all of the analysed muscles had the highest activity during kettlebell pressing with a weight of 70% of 1RM. The largest differences between trials of 70% of 1RM pressing kettlebell and dumbbell of 29% and 24.2% were noted for the serratus anterior and lower trapezius, respectively. The remaining anterior deltoid, posterior deltoid, spinal erector, and upper trapezius muscles had 19%, 7.5%, 4% and 1.5% higher activities for kettlebell pressing, respectively. Moreover, it is worth mentioning that during the 6 kg pressing, higher muscle activities were also recorded for kettlebells, successively for the serratus anterior (18.1%), lower trapezius (10%), posterior deltoid (3.6%), spinal erector (1.2%), and anterior deltoid (0.88%). This relation was not observed for upper trapezius muscle, where there was 3.4% higher activity during the 6 kg dumbbell press.

Such tendency may be influenced by the placement of the kettlebell and the dumbbell’s center of gravity [8]. The kettlebell’s center of gravity is located below the grip, within the ball. This can affect the movement due to additional rotational torque, which requires higher muscle activity to balance this torque during press movement. Only one paper similar to the present study was found. Dicus et al. [8] examined EMG activity of the anterior deltoid and pectoralis major muscle activity when performing an overhead press in the abduction position of the upper arm with 25% of 1RM kettlebell and dumbbell. They showed that the dumbbell press requires significantly more muscle activation during the exercise. These results stay the opposite to the results of the present study, which showed higher activation during kettlebell exercises with 70%1RM load in five out of six assessed muscles (except upper trapezius). Differences can be caused by higher load and different movement directions in our study, which can influence muscle activity.

Büll et al. [14] assessed the activity of the trapezius and serratus anterior muscles during the barbell overhead press in sitting and standing with a narrow and wide grip. Muscle activity increased during each movement, but no differences were visible between trials. This can be caused by low-load exercises with a light wooden stick instead of a real barbell. Therefore, the cited study does not provide information on how higher loads can influence muscle activity in different exercise modifications. The overhead press in the sagittal plane (with the elbow joint facing forward) was assessed by Andersen et al. [9]. In their study, the overhead press was performed with different intensities (Borg CR10 levels 3 and 8). It was noticed that exercises with higher intensity result in higher activity of the serratus anterior and upper trapezius muscles, whereas the activation of the lower trapezius was less affected. In the present study higher activity of these muscles was noted with higher loads, although statistically significant differences were found for lower trapezius when comparing dumbbell and kettlebell pressing with 6 kg load to kettlebell pressing with 70% of 1RM load. Upper trapezius was significantly less active during kettlebell pressing with 6 kg load versus dumbbell and kettlebell pressing with 70% of 1RM load. Serratus anterior muscle was significantly less active during dumbbell pressing with 6 kg load versus dumbbell and kettlebell pressing with 70% of 1RM load. The activation of the upper trapezius was different than other muscles. It is worth mentioning that the mean load in the present study was almost double that used in the study of Andersen et al. [9], which may affect the results. Additionally, they included only women in the study, while our study was based on men. Different body constitutions, as well as thorax and shoulder structure proportions, can influence the results obtained in these two studies [18].

Comparable load (70% of 1RM) in dumbbells overhead pressing in a sitting position was analysed by Paol et al. [7]. They assessed the effect of increased range of motion (RoM) at different loads on the activity of eight muscles (clavicular head of pectoralis major, anterior, medium, and posterior deltoid, upper and middle trapezius, long head of triceps, and teres minor). Different RoM consisted of final elbow angle at the level of 90° (R1), 135° (R2), and 180° (R3). The authors showed that the use of the widest RoM increased the activity of all selected muscles with respect to the closest one. Moreover, EMG activity of all muscles increased, as expected, in rough proportion with the increase of the load, which was in line with our study. Paol et al. [7] showed that the trapezius and the deltoid muscles were the most involved muscles in this exercise with each of the 3 ROMs, confirming the previous reports by Büll et al. [14] on trapezius activity. It is worth adding that in this paper the study group consisted of only six men with at least 3 years of experience in strength training. Schick et al. [28] examined pectoralis major, anterior and medial deltoid activation during a Smith machine and free weight bench press at lower (70% of 1RM) and higher (90% of 1RM) intensities. They found greater activation only of the medial deltoid on the free-weight bench press than on the Smith machine bench press. They also noted greater muscle activation at 90% of 1RM than at 70% of 1RM load. Moreover, they did not find differences in muscle activation for the anterior deltoid and pectoralis major between experience levels. Saeterbakken and Fimland [10] examined the effect of performing upper-body resistance exercises with dumbbells versus barbells and standing versus seated. They measured EMG muscle activity of anterior, medial, and posterior deltoids, biceps and triceps brachii. The authors noted higher activity of the deltoid muscle, both anterior and posterior parts, during dumbbell overhead press in a standing position than in barbell overhead press. Additionally, the load in dumbbell pressing was lower than in exercise with a barbell. In contrast, Kohler et al. [11] noticed a similar activity of the deltoid muscle in barbell and dumbbell pressing. These varying findings might be the result of comparisons made based on unmatched loads and different positions during pressing. In a standing position, the body has more degrees of freedom. This result in higher activation of the core stabilizing muscles, which can also result in higher activity in local stabilizers. The other reason can be the fact that the standing position requires the activity of the trunk and lower limb muscles, which affects higher co-contraction of the muscles in the whole kinematic chain to maintain body balance [29]. Williams, et al. [17] showed higher spinal erector muscle activity in standing overhead press using 50% of 1RM load on a barbell with kettlebells hanging on elastic bands than with static load. Unstable load and differences in the center of mass position probably increased trunk muscle activation to maintain shoulder stability. On the other hand, Kohler, et al. [11] noticed higher activity of the spinal erector muscle during barbell pressing in sitting on a Swiss ball. Barbell press moves the center of mass of the equipment forward when compared to dumbbell pressing, which can affect the activation of back muscles operating on a longer lever. Additionally, an unstable base (like a Swiss ball) can increase back muscle activation [30]. The last problem worth discussing concerns the unilateral performance of the dumbbell/ kettlebell press. It is worth emphasizing the fact that such a position increases stabilization function, which has been shown in hip resistance exercises [31]. In addition, one-handed pressing provides the purity of movement performance, since there is no need to evaluate the symmetry of muscle action and possible compensatory functions on the part of the other limb.

Some limitations of this study have to be acknowledged. The study group consisted of young, athlete males. Most of them were preparing to complete Kettlebell instructor training or were enthusiasts of this type of training. Therefore, the results may not reflect the other population groups, even if the effect size came out at a medium to high level. No maximum voluntary contraction was assessed in this study, which makes the results of different participants not comparable with each other. In future studies, more individuals should be investigated; exercise trials should be assessed with EMG synchronized with video or 3D movement analysis to assess muscles’ activation patterns, including timing. It would be worthful to include more shoulder and core muscles in future studies, as well as to analyse different techniques of the overhead press.

## 5. Conclusions

This study is the first comparing muscle activation during the overhead press in the position of the upper limb in adduction with a kettlebell and a dumbbell with a high load. The results showed that kettlebell overhead pressing both with 6 kg and 70% of 1RM load resulted in higher activity in five out of six assessed muscles, including anterior and posterior deltoid, lower trapezius, serratus anterior and erector spinae in comparison to activation obtained during the equal weight of dumbbells. It is worth highlighting that, although there were no statistically significant differences between the muscle activity recorded for lifting kettlebells and dumbbells of the same weight, the activation of the analyzed muscles was at any time higher during the kettlebell press task. Different center of gravity locations in the kettlebell versus the dumbbell can increase shoulder muscle activity during the overhead press.

Thus, the prevailing opinion in gyms that kettlebell training is an alternative in the final stages of improvement, especially the development of strength, mobility, power, dynamic stability and cardiorespiratory fitness is accurate. The result of this study shows that kettlebell exercises are more effective, which again suggests that it is worthwhile to supplement standard training routines with kettlebell exercises. However, more studies are required to confirm these results.

## Figures and Tables

**Figure 1 sensors-22-09762-f001:**
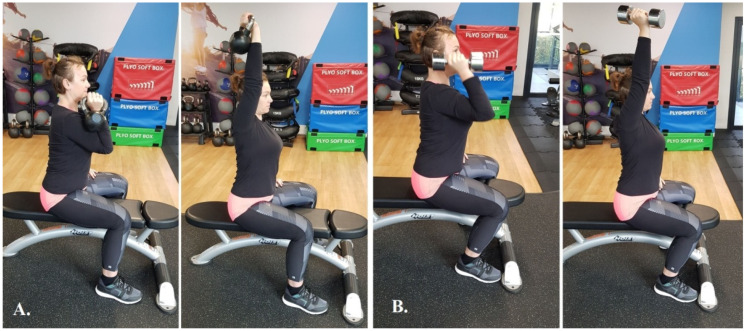
Kettlebell and dumbbell overhead press. (**A**) Starting and final position for kettlebell overhead presses; (**B**) starting and final position for dumbbell overhead presses.

**Figure 2 sensors-22-09762-f002:**
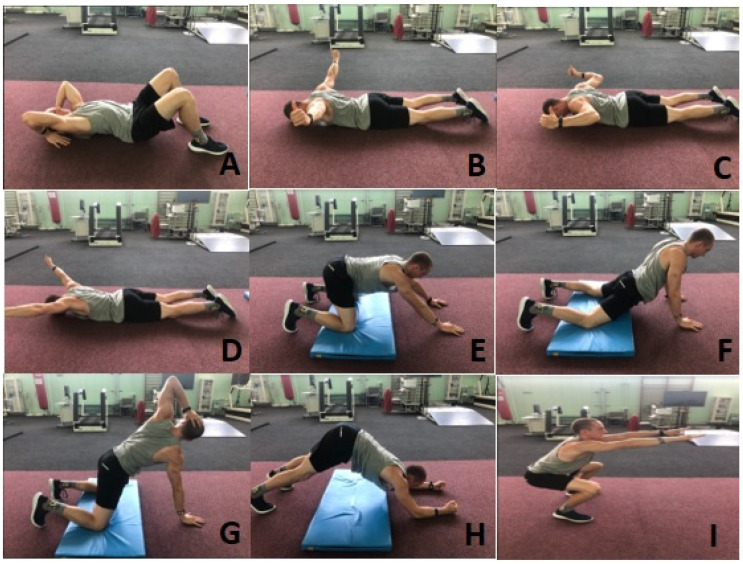
Warm-up protocol. (**A**–**D**) Mobilization of the upper limb joints; (**E**,**F**) mobilization of the hip joints; (**G**,**H**) mobilization of the thoracic spine; (**I**) squat without loading.

**Figure 3 sensors-22-09762-f003:**
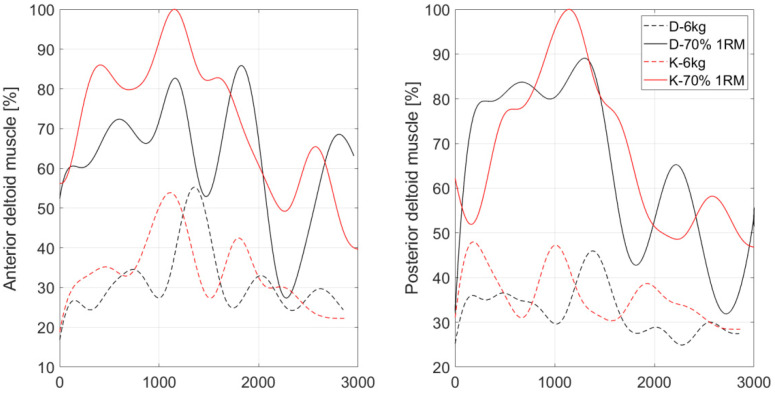
Example waveform of normalized muscle activity for anterior deltoid and posterior deltoid muscles, where: D-6kg—6 kg dumbbell press, K-6kg—6 kg kettlebell press, D-70% 1RM—70% of the 1RM dumbbell press, K-70% 1RM—70% of the 1RM kettlebell press.

**Figure 4 sensors-22-09762-f004:**
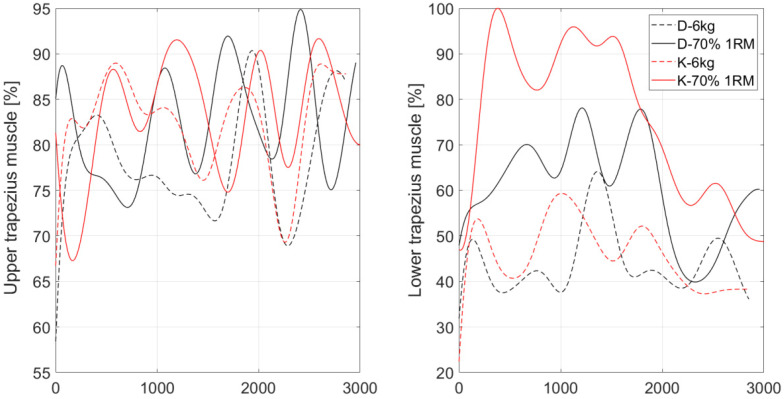
Example waveform of normalized muscle activity for upper and lower trapezius muscles, where: D-6kg—6 kg dumbbell press, K-6kg—6 kg kettlebell press, D-70% 1RM—70% of the 1RM dumbbell press, K-70% 1RM—70% of the 1RM kettlebell press.

**Figure 5 sensors-22-09762-f005:**
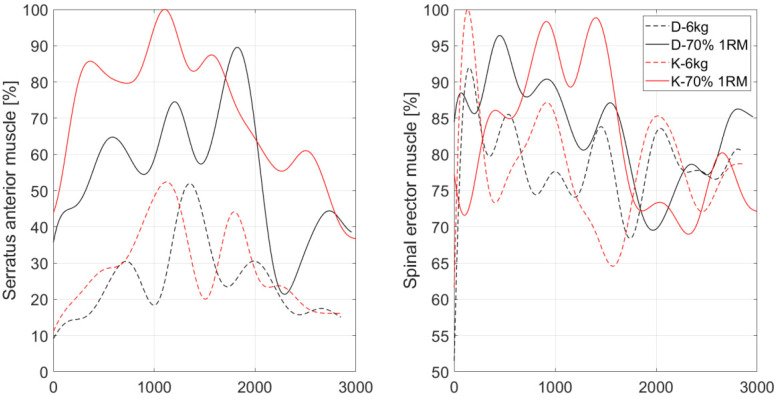
Example waveform of normalized muscle activity for serratus anterior and spinal erector muscles, where: D-6kg—6 kg dumbbell press, K-6kg—6 kg kettlebell press, D-70% 1RM—70% of the 1RM dumbbell press, K-70% 1RM—70% of the 1RM kettlebell press.

**Table 1 sensors-22-09762-t001:** Participants’ characteristics (mean ± SD).

Group	Age [Years]	Body Mass [kg]	Body Height [cm]	Experience with Resistance Training [Months]	70% 1RM Kettlebell Overhead Press	70% 1RM Dumbbell Overhead Press
N = 20 (male)	24.9 ± 2	85.5 ± 38.4	181.8 ± 6.5	19 ± 4.04	17.5 ± 2.8	18.3 ± 2.7

Abbreviations: 70% 1RM = 70% of one repetition maximum.

**Table 2 sensors-22-09762-t002:** Median, lower and upper quartile (Q1; Q3) of the maximum activity of the anterior part of muscle deltoid and posterior part of muscle deltoid, where: D-6kg—6 kg dumbbell press, K-6kg—6 kg kettlebell press, D-70% 1RM—70% of the 1RM dumbbell press, K-70% 1RM—70% of the 1RM kettlebell press. The *p*-values are for the Dunn Bonferroni post-hoc test.

D-70% 1RM	D-6kg	K-70% 1RM	K-6kg	*p*-Value
**Anterior deltoid muscle—Median [Q1; Q3]**				
84 [77.25; 100]	56.5 [41; 62.25]	100 [77; 100]	57 [41.5; 74.5]	D-70%1RM vs. D-6kg (*p* = 0.0059) D-70%1RM vs. K-6kg (*p* = 0.0405) K-70%1RM vs. D-6kg (*p* = 0.0003) K-70%1RM vs. K-6kg (*p* = 0.0034)
**Posterior deltoid muscle—Median [Q1; Q3]**				
93 [81.25; 100]	55 [44.5; 62]	100 [85.75; 100]	56 [44.5; 71.75]	D-70%1RM vs. D-6kg (*p* = 0.0026) D-70%1RM vs. K-6kg (*p* = 0.0204) K-70%1RM vs. D-6kg (*p* = 0.0002) K-70%1RM vs. K-6kg (*p* = 0.0026)

**Table 3 sensors-22-09762-t003:** Median, lower and upper quartile (Q1; QQ) of the maximum activity of the upper trapezius muscle and lower trapezius muscle, where: D-6kg—6 kg dumbbell press, K-6kg—6 kg kettlebell press, D-70% 1RM—70% of the 1RM dumbbell press, K-70% 1RM—70% of the 1RM kettlebell press. The *p*-values are for the Dunn Bonferroni post-hoc test.

D-70% 1RM	D-6kg	K-70% 1RM	K-6kg	*p*-Value
**Upper trapezius muscle—Median [Q1; Q3]**				
96.5 [91.25; 100]	90.5 [68; 97]	98 [92.25; 100]	87.5 [54.5; 93.75]	D-70%1RM vs. K-6kg (*p* = 0.0126) K-70%1RM vs. K-6kg (*p* = 0.0204)
**Lower trapezius muscle—Median [Q1; Q3]**				
80.5 [65.75; 89]	64.5 [52.25; 72.75]	100 [90.25; 100]	71 [54; 83.75]	K-70%1RM vs. D-6kg (*p* = 0.0001) K-70%1RM vs. K-6kg (*p* = 0.0076)

**Table 4 sensors-22-09762-t004:** Median, lower and upper quartile [Q1; Q3] of the maximum activity of the serratus anterior and spinal erector- thoracis part, where: D-6kg—6 kg dumbbell press, K-6kg—6 kg kettlebell press, D-70% 1RM—70% of the 1RM dumbbell press, K-70% 1RM—70% of the 1RM kettlebell press. The *p*-values are for the Dunn Bonferroni post-hoc test.

D-70% 1RM	D-6kg	K-70% 1RM	K-6kg	*p*-Value
**Serratus anterior muscle—Median [Q1; Q3]**				
77 [70; 88.25]	58 [37.5; 66.25]	100 [95.5; 100]	68.5 [51.25; 79.75]	D-70%1RM vs. D-6kg (*p* = 0.0204) K-70%1RM vs. D-6kg (*p* = 0.0001)
**Spinal erector- thoracis muscle—Median [Q1; Q3]**				
95.5 [80.5; 99.5]	80.5 [69.25; 91.25]	100 [95; 100]	81.5 [72.5; 88.25]	D-70%1RM vs. D-6kg (*p* = 0.0045) D-70%1RM vs. K-6kg (*p* = 0.0045) K-70%1RM vs. D-6kg (*p* = 0.0001) K-70%1RM vs. K-6kg (*p* = 0.0001)

## Data Availability

The measurement data used to support the findings of this study are available from the corresponding author upon request.

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
