# Peer review of "The Effect of the Weight and Type of Equipment on Shoulder and Back Muscle Activity in Surface Electromyography during the Overhead Press—Preliminary Report"

_sensors, 2022, doi:10.3390/s22249762_

Round 1

Reviewer 1 Report

This study compared the 12 activity of shoulder and back muscles during the overhead press. The research is interesting. I have some suggestions as follow. 

[1] The temporal display of 12 muscle activity should be added to compare the difference. 

[2] What is the practical research value of this study?  

[3] The format of references is not uniform.

Author Response

Dear Reviewer,

Thank you for taking the time to read the manuscript and give us valuable comments. We tried to address all of them. We hope, that the corrected version meets your requirements and will be accepted for publication.

This study compared the 12 activity of shoulder and back muscles during the overhead press. The research is interesting. I have some suggestions as follow.

[1] The temporal display of 12 muscle activity should be added to compare the difference.

Thank you for this comment, EMG signal waveforms have been added in the paper. We only added illustrative graphs because, unfortunately, they do not turn on and reach maximum values at similar times in all individuals. This makes it impossible to present them in an aesthetically pleasing way like for example for gait kinematic parameters. It is worth mentioning that no study analyzing pressing motions has presented time courses of muscle activity.

[2] What is the practical research value of this study?

The opinion in the gyms is that kettlebell training is an alternative in the final stages of improvement, especially the development of strength, mobility, power, stabilization and cardiorespiratory fitness. The kettlebell weight is built in such a way that it has a shifted center of gravity, since its mass is not evenly distributed around the handle, unlike that of a dumbbell or barbell. This forces the trainee to control the weight, provoking the tightening of many muscles throughout the body. Equipment with such properties will be ideal for stability work, and what has been proven to work during exercise increases the activity of the analyzed muscles. Such a result of our study shows that kettlebell exercises are more effective, which again suggests that it is worth supplementing standard training routines with kettlebell exercises.

The above-mentioned aspects were included in the manuscript in the Introduction, Discussion and Conclusions.

 [3] The format of references is not uniform.

Thank you for this comment. The paper uses the EndNote program, and the style was downloaded from the MDPI journal website. We tried to correct all mistakes manually. If anything is still missing, we will correct it before publication according to Journals Editors comments.

Reviewer 2 Report

Although the methodology of the EMG study is correct, the way the data is presented does not add anything new. The study group was too small to describe such significant findings. In addition, the authors did not calculate and show data related to the effect size and the sample size, so the social dimension of the study decreases the value (these are basic statistics in science today).

Author Response

Dear Reviewer,

Thank you for taking the time to read the manuscript and give us valuable comments. We hope, that the corrected version meets your requirements and will be accepted for publication.

Although the methodology of the EMG study is correct, the way the data is presented does not add anything new. The study group was too small to describe such significant findings. In addition, the authors did not calculate and show data related to the effect size and the sample size, so the social dimension of the study decreases the value (these are basic statistics in science today).

Thank you for this comment. The paper has been corrected and completed with the details listed. In most cases, the effect size is moderate or large.

Reviewer 3 Report

Below are my comments

Introduction

Overall, I think the introduction is well written and makes a great case for why this work needs to be examined. 

Methods

The study needs more clarity in the methodology to help improve replicability

1. How were subjects recruited? Where were they recruited from? 

2. Please present the mean +/- SD for number of months of experience with resistance training

3. How was ROM in shoulder and elbow assessed?

4. I am assuming that all participants were right handed? I would just make sure you put that in your methodology in the participants section as well

5. You might want to put the 1-RM protocol before the procedures because I was wondering how 1-RM was assessed when you were talking about using 70% of 1RM for your protocol.

6. Was the 1RM assessed on a different day? 

7. How many minutes rest did the participants get after each exercise?

8. The incorrect statistical analysis test was completed. Since this is a within subject design and your participants are basically completing all 4 measures, a Kruskal Wallis will measure group differences, while a Friedman test is more in line with a repeated measures ANOVA (which is basically what your study design calls for). With that being said, a Wilcoxan-Rank sum test with a Bonferroni adjustment will need to be applied. Considering the number of post-hoc analyses that would need to be conducted you will need to adjust your p-value. The Dunn test is best for between group differences, which your study does not have. 

Results

Although I cannot determine the validity of the results since the incorrect statistical analysis was completed, one of things that I would make sure is that the authors use their results and have some sort of table where post-hoc results can be easily displayed for each significant analysis

Author Response

Dear Reviewer,

Thank you for taking the time to read the manuscript and give us valuable comments. We tried to address all of them. We hope, that the corrected version meets your requirements and will be accepted for publication.

Introduction

Overall, I think the introduction is well written and makes a great case for why this work needs to be examined.

Thank you for this positive comment.

Methods

The study needs more clarity in the methodology to help improve replicability.

  1. How were subjects recruited? Where were they recruited from?

Participants were recruited from gyms where one of the authors (MB) is a fitness instructor. These included individuals preparing to complete Kettlebell instructor training or enthusiasts of this type of training.

  1. Please present the mean +/- SD for number of months of experience with resistance training.

Thank you for your comment. This part of the paper has been corrected.

  1. How was ROM in shoulder and elbow assessed?

The ROM in the shoulder and elbow was assessed using goniometer. For the shoulder joint, the subject performed active motions in all functional planes, which included flexion, extension, extension, adduction, adduction, and internal and external rotation [Stubbs et al.]. For the elbow joint, the individuals performed flexion, extension, pronation and supination movements according to typical procedure described by Zwerus et al. Appropriate information was added to the manuscript.

Stubbs, N.; Fernandez, J.; Glenn, W. Normative data on joint ranges of motion of 25- to 54-year-old males. International Journal of Industrial Ergonomics - INT J IND ERGONOMIC 1993, 12, 265-272, doi:10.1016/0169-8141(93)90096-V.
Zwerus, E.L.; Willigenburg, N.W.; Scholtes, V.A.; Somford, M.P.; Eygendaal, D.; van den Bekerom, M.P. Normative values and affecting factors for the elbow range of motion. Shoulder & elbow 2019, 11, 215-224, doi:10.1177/1758573217728711.
  1. I am assuming that all participants were right handed? I would just make sure you put that in your methodology in the participants section as well.

Yes, the subjects were right-handed. This information was previously included in the text in the protocol section. Now this information is also in the participants’ characteristic section.

  1. You might want to put the 1-RM protocol before the procedures because I was wondering how 1-RM was assessed when you were talking about using 70% of 1RM for your protocol.

Thank you, the manuscript has been corrected as suggested.

  1. Was the 1RM assessed on a different day?

Yes, the information was included in the text, but now it is more specific. The procedure was performed by each participant in the week preceding the measurements: “Seven and four days before the study, 1RM with kettlebell and 1RM with dumbbell were determined, respectively”.

  1. How many minutes rest did the participants get after each exercise?

Each individual rested for about 5 minutes between exercises with a 1RM load, and for a series with a 6kg load, about 5 minutes.

  1. The incorrect statistical analysis test was completed. Since this is a within subject design and your participants are basically completing all 4 measures, a Kruskal Wallis will measure group differences, while a Friedman test is more in line with a repeated measures ANOVA (which is basically what your study design calls for). With that being said, a Wilcoxan-Rank sum test with a Bonferroni adjustment will need to be applied. Considering the number of post-hoc analyses that would need to be conducted you will need to adjust your p-value. The Dunn test is best for between group differences, which your study does not have.

Thank you for this comment. It was our mistake. Statistical analysis has been corrected, but the results are very similar.

Results

Although I cannot determine the validity of the results since the incorrect statistical analysis was completed, one of things that I would make sure is that the authors use their results and have some sort of table where post-hoc results can be easily displayed for each significant analysis.

Thank you for this comment, this section has indeed been corrected.

Reviewer 4 Report

The manuscript „The effect of the weight and type of equipment on shoulder and back muscle activity in surface electromyography during the overhead press” is novel, and fills out the research gap in resistance training exercises. The introduction includes sufficient justification for the study’s aim and provides relevant references. I missed only the brief description of overhead presses to the bench pressing.

The method section allows the study replication and is of appropriate design. However, there should be a much more specific justification of the EMG  normalization procedure and the speed of muscle action during the experiment.

The result section is clear and easy to follow, however, the figures should be of better quality. The axis lines should be thicker and no gridlines are necessary.

The discussion is relevant and includes enough study consequences. However, it should be noted possible gender difference, performance level differences and justification of only unilateral condition to be studied.

Line 21 – 23: Please be more specific for the conclusion. This conclusion is too general.

Line 24: Some keywords are already in the title, please use more general ones, e.g.; resistance exercise.

Line 27 – 45: The first paragraph describes what muscle contributions are relevant for overhead presses, but this paragraph should also present how overhead press differs from other pressing exercises e.g. bench press.

https://www.gymnica.upol.cz/artkey/gym-201804-0001_muscle_activity_during_the_incline_shoulder_press_in_relation_to_the_exercise_intensity.php

Line 106 – 112:

Does gender influences the EMG, during pressing exercises? Make note of the men selection in the discussion or method part.

https://www.ncbi.nlm.nih.gov/pmc/articles/PMC6234306/

Line 138 – 143: What tempo did you use and why?

https://www.ncbi.nlm.nih.gov/pmc/articles/PMC7075231/

Line 179: The selection of normalization method should be  justified by recent recommendations.

 Besomi, M., Hodges, P. W., Clancy, E. A., Van Dieën, J., Hug, F., Lowery, M., ... & Tucker, K. (2020). Consensus for experimental design in electromyography (CEDE) project: Amplitude normalization matrix. Journal of Electromyography and Kinesiology53, 102438.

Line 338:

Why did you select the unilateral performance of the dummbell/Kettlebell press? Not synchronized lift with right and left load. You might discuss that your point was to increase the stabilization functions, which has been shown in hip joint resistance exercises.

https://www.ncbi.nlm.nih.gov/pmc/articles/PMC4640053/

Author Response

Dear Reviewer,

Thank you for taking the time to read the manuscript and give us valuable comments. We tried to address all of them. We hope, that the corrected version meets your requirements and will be accepted for publication.

The manuscript „The effect of the weight and type of equipment on shoulder and back muscle activity in surface electromyography during the overhead press” is novel, and fills out the research gap in resistance training exercises. The introduction includes sufficient justification for the study’s aim and provides relevant references. I missed only the brief description of overhead presses to the bench pressing.

Thank you, the following information was included in the manuscript: “In strength sports, the primary motion that engages the muscles of the upper limbs are different types of pressing motion [1,2]. The two most popular pressing motions are bench and overhead. They differ not only in the alignment of body segments to each other but, before anything else, in the activity of the muscles involved in a given motion. The bench press is performed in the lying position, and the overhead press is in the standing or sitting position. This changes where the loading is relative to the spine - perpendicular for the bench press and inline for the overhead press. According to Stronska, Bojacz, Golas, Maszczyk, Zajac and Stastny [1], the bench press uses greater activation of the anterior deltoid and pectorals major muscles than the lateral head triceps brachii and long head triceps brachii at all exercise intensities (40% 1RM, 60% 1RM, 80% 1RM and 100% 1RM).”

The method section allows the study replication and is of appropriate design. However, there should be a much more specific justification of the EMG normalization procedure and the speed of muscle action during the experiment.

Thank you for this comment, this information was added to the method section.

The result section is clear and easy to follow, however, the figures should be of better quality. The axis lines should be thicker and no gridlines are necessary.

Thank you, we changed this section. Tables were introduced instead of figures, at the request of another of the Reviewers. We did not want to duplicate values from tables in figures.

The discussion is relevant and includes enough study consequences. However, it should be noted possible gender difference, performance level differences and justification of only unilateral condition to be studied.

Thank you for this comment. The manuscript has been partly rewritten according to your comments below.

Line 21 – 23: Please be more specific for the conclusion. This conclusion is too general.

Thank you, the paper has been improved according to your comment.

Line 24: Some keywords are already in the title, please use more general ones, e.g.; resistance exercise.

Thank you, we have changed the keywords as suggested.

Line 27 – 45: The first paragraph describes what muscle contributions are relevant for overhead presses, but this paragraph should also present how overhead press differs from other pressing exercises e.g. bench press. https://www.gymnica.upol.cz/artkey/gym-201804-0001_muscle_activity_during_the_incline_shoulder_press_in_relation_to_the_exercise_intensity.php

Thank you for this comment. The paper has been updated with this information.

 Line 106 – 112: Does gender influences the EMG, during pressing exercises? Make note of the men selection in the discussion or method part. https://www.ncbi.nlm.nih.gov/pmc/articles/PMC6234306/

Thank you for this comment. The paper has been updated with this information.

Line 138 – 143: What tempo did you use and why? https://www.ncbi.nlm.nih.gov/pmc/articles/PMC7075231/

Thank you for this comment. The paper has been updated with this information. “Each participant performed the trials twice in random order with the volitional cadence. The motion cadence was not specified because it changed the execution of the exercise to artificial (robotic). However, the timing of motion was assessed with a watch and averaged 3sec/0/2sec for all individuals”

Line 179: The selection of normalization method should be justified by recent recommendations.  Besomi, M., Hodges, P. W., Clancy, E. A., Van Dieën, J., Hug, F., Lowery, M., ... & Tucker, K. (2020). Consensus for experimental design in electromyography (CEDE) project: Amplitude normalization matrix. Journal of Electromyography and Kinesiology53, 102438.

Thank you for this comment. The manuscript was changed according to your suggestion.

Line 338: Why did you select the unilateral performance of the dummbell/Kettlebell press? Not synchronized lift with right and left load. You might discuss that your point was to increase the stabilization functions, which has been shown in hip joint resistance exercises. https://www.ncbi.nlm.nih.gov/pmc/articles/PMC4640053/

Thank you, this part was completed in the discussion.

Round 2

Reviewer 2 Report

How is the purpose of the study (poorly defined) supposed to support the training process when the results are irrelevant and a small group is surveyed. What is the applied value of the findings - "The present study may help to improve the training of people who want to increase the activation and the strength of the upper limb girdle muscles."  

The authors still have not described on what basis they determined the size of the group (N=20), which in my opinion is too small to make inferences on this topic. They also did not describe what they want to achieve by obtaining the analysed effect size - no interpretation. 

According to the description shown, it is not possible to repeat the study without making a mistake. 

In addition, the POWER of the test is still not described - which demonstrates the effectiveness of repeating the study. !

When examining an amateur group, many people can be verified for such an inference.

Author Response

Dear Reviewer,

Thank you for taking the time to read the manuscript and give us valuable comments. We tried to address all of them. We hope, that the corrected version meets your requirements and will be accepted for publication.

How is the purpose of the study (poorly defined) supposed to support the training process when the results are irrelevant and a small group is surveyed. What is the applied value of the findings - "The present study may help to improve the training of people who want to increase the activation and the strength of the upper limb girdle muscles."

Thank you for this comment. The cited text was deleted from the paper after the first review and the 2nd version of the manuscript did not include it.

This study aimed to compare the activity of shoulder and back muscles, measured by surface electromyography, during the overhead press in the sagittal plane in a sitting position with a kettlebell and a dumbbell. The opinion in the gyms is that kettlebell training is an alternative in the final stages of improvement, especially the development of strength, mobility, power, stabilization and cardiorespiratory fitness. The kettlebell weight is built in such a way that it has a shifted center of gravity, since its mass is not evenly distributed around the handle, unlike that of a dumbbell or barbell. This forces the trainee to control the weight, provoking the tightening of many muscles throughout the body. Equipment with such properties will be ideal for stability work, and what has been proven to work during exercise increases the activity of the analyzed muscles. Such a result of our study shows that kettlebell exercises are more effective, which again suggests that it is worth supplementing standard training routines with kettlebell exercises.

The above-mentioned aspects were included already in the 2nd version of the manuscript in the Introduction, Discussion and Conclusions.

Additionally, it has been pointed out in the study limitations what should be completed and what is perhaps disputed in the research.

The authors still have not described on what basis they determined the size of the group (N=20), which in my opinion is too small to make inferences on this topic.

Thank you for this comment. The following are listed papers in the field of the topic undertaken, which were cited in this paper. We have excluded papers that dealt with other topics or reviews. As you can see, most of the papers (13 out of 19) have a much smaller number of subjects studied. We also want to emphasize that the people studied were not typical amateurs who saw a kettlebell for the first time. They were people preparing to become instructors, and there aren't many of those. So in this aspect, the group is representative. We could not undertake an evaluation of people who do not technically perform the motion.

Apart from this, other reviewers reported, that the sample size is relevant. Since there is a lot of doubt about the number of subjects in your comments, we decided to emphasize this fact, showing that this is a preliminary report. A small number of participants was also discussed in the study limitations.

Paper

Number of participants

Stronska, K.; Bojacz, P.; Golas, A.; Maszczyk, A.; Zajac, A.; Stastny, P. Muscle activity during the incline shoulder press in relation to the exercise intensity. Acta Gymnica 2018, 48, 141-146.

8

Maszczyk, A.; Wilk, M.; Krzysztofik, M.; Gepfert, M.; ZajÄ…c, A.; Petr, M.; Stastny, P. The effects of resistance training experience on movement characteristics in the bench press exercise. Biol. Sport 2020, 37, 79-83.

32 novice men + 36 advanced group

Ichihashi, N.; Ibuki, S.; Otsuka, N.; Takashima, S.; Matsumura, A. Kinematic characteristics of the scapula and clavicle during military press exercise and shoulder flexion. J Shoulder Elbow surg 2014, 23, 649-657.

16

Riek, L.M.; Tome, J.; Ludewig, P.M.; Nawoczenski, D.A. Improving Shoulder Kinematics in Individuals With Paraplegia: Comparison Across Circuit Resistance Training Exercises and Modifications in Hand Position. Phys Ther 2016, 96, 1006-1017.

18

Paoli, A.; Marcolin, G.; Petrone, N. Influence of different ranges of motion on selective recruitment of shoulder muscles in the sitting military press: an electromyographic study. J Strength Cond Res 2010, 24, 1578-1583.

6

Dicus, J.R.; Holmstrup, M.E.; Shuler, K.T.; Rice, T.T.; Raybuck, S.D.; Siddons, C.A. Stability of Resistance Training Implement alters EMG Activity during the Overhead Press. Int. J. Exerc. Sci. 2018, 11, 708-716.

21

Andersen, C.H.; Zebis, M.K.; Saervoll, C.; Sundstrup, E.; Jakobsen, M.D.; Sjøgaard, G.; Andersen, L.L. Scapular muscle activity from selected strengthening exercises performed at low and high intensities. J Strength Cond Res 2012, 26, 2408-2416.

17

Saeterbakken, A.H.; Fimland, M.S. Effects of body position and loading modality on muscle activity and strength in shoulder presses. J Strength Cond Res 2013, 27, 1824-1831.

15

Kohler, J.M.; Flanagan, S.P.; Whiting, W.C. Muscle activation patterns while lifting stable and unstable loads on stable and unstable surfaces. J Strength Cond Res 2010, 24, 313-321.

30

Saeterbakken, A.H.; Stien, N.; Pedersen, H.; Solstad, T.E.J.; Cumming, K.T.; Andersen, V. The Effect of Grip Width on Muscle Strength and Electromyographic Activity in Bench Press among Novice- and Resistance-Trained Men. Int. J. Environ. Res. Public Health 2021, 18.

15 - the resistance-trained group and 13 - the novice-trained group

Büll, M.L.; Vitti, M.; Freitas, V.; Rosa, G.J. Electromyographic validation of the trapezius and serratus anterior muscles in military press exercises with open and middle grip. Electromyogr Clin Neurophysiol 2001, 41, 203-207.

24

Williams, M.; Hendricks, D.; Dannen, M.; Arnold, A.; Lawrence, M. Activity of Shoulder Stabilizers and Prime Movers During an Unstable Overhead Press. J Strength Cond Res 2018, 34, 1.

12

Gołaś, A.; Maszczyk, A.; Pietraszewski, P.; Wilk, M.; Stastny, P.; Strońska, K.; Studencki, M.; Zając, A. Muscular activity patterns of female and male athletes during the flat bench press. Biol. Sport 2018, 35, 175-179.

10

Chen, H.T.; Wu, H.J.; Chen, Y.J.; Ho, S.Y.; Chung, Y.C. Effects of 8-week kettlebell training on body composition, muscle strength, pulmonary function, and chronic low-grade inflammation in elderly women with sarcopenia. Exp. Gerontol. 2018, 112, 112-118.

33

García-Ramos, A.; Suzovic, D.; Pérez-Castilla, A. The load-velocity profiles of three upper-body pushing exercises in men and women. Sports Biomech. 2021, 20, 693-705.

12 men and 12 women

Schick, E.E.; Coburn, J.W.; Brown, L.E.; Judelson, D.A.; Khamoui, A.V.; Tran, T.T.; Uribe, B.P. A comparison of muscle activation between a Smith machine and free weight bench press. J Strength Cond Res 2010, 24, 779-784.

14 experienced and 12 inexperienced

Ebben, W.P.; Petushek, E.J.; Fauth, M.L.; Garceau, L.R. EMG analysis of concurrent activation potentiation. Med Sci Sports Exerc. 2010, 42, 556-562.

11 men and 12 women

Kang, H.; Jung, J.; Yu, J. Comparison of trunk muscle activity during bridging exercises using a sling in patients with low back pain. J Sports Sci Med 2012, 11, 510-515.

30

Stastny, P.; Lehnert, M.; Zaatar, A.M.; Svoboda, Z.; Xaverova, Z. Does the Dumbbell-Carrying Position Change the Muscle Activity in Split Squats and Walking Lunges? J Strength Cond Res 2015, 29, 3177-3187.

14 resistance trained group and 14 non resistance trained group.

They also did not describe what they want to achieve by obtaining the analysed effect size - no interpretation.

Effect size was calculated because you asked for it in the first review. The effect size was medium and large. Now, an additional interpretation has been added also to the discussion. Of course, it is known that the effect size tells you how meaningful the relationship between variables or the difference between groups is. It indicates the practical significance of a research outcome. A large effect size means that a research finding has practical significance, while a small effect size indicates limited practical applications.

According to the description shown, it is not possible to repeat the study without making a mistake.

Thank you for this comment. The other Reviewers did not report this problem. We do not know where we should improve the description of how the measurement was performed. We would appreciate it if you could clarify which part is not understood.

In addition, the POWER of the test is still not described - which demonstrates the effectiveness of repeating the study! When examining an amateur group, many people can be verified for such an inference.

Power has been calculated in PQStat in base on means and standard deviations and it was mostly >0.90, apart of upper trapezius muscle, where the power was 0.86. Please remember that these were not typical amateurs. These were people who were preparing to complete the Kettlebell instructor course. At this point most of them are no longer amateurs. Quoting the definition of an amateur, we could not write otherwise. An amateur is 1.1 someone who is engaged in something reliably or does some work without professional training; 1.2 an athlete who does sports only for pleasure, without material gain. The selection of such a group of people guaranteed us the correct motion technique, which is crucial in such studies.

Reviewer 3 Report

While I appreciate the author addressing many of my concerns, it seems that the authors are still using between subjects post-hoc analyses instead of within subjects. Please confer with a statistician regarding your statistical analyses. 

Author Response

Dear Reviewer,

Thank you for taking the time to read the manuscript and give us valuable comments.

While I appreciate the author addressing many of my concerns, it seems that the authors are still using between subjects post-hoc analyses instead of within subjects. Please confer with a statistician regarding your statistical analyses.

Dunn test with Bonferroni correction for multiple comparisons (sometimes called Dunn-Bonferroni test) is one of the post-hoc tests used for intra-group comparisons [Dunn, 1964]. Other post-hoc tests used in the within-subjects analysis are Dunn-Sidak or Conover-Inman tests. Dunn-Bonferroni test was used several times in medical literature [Dreher et al., 2020; Casanova-Alvarez et al. 2021; Xu et al., 2021], as well as in our previously published articles [Hadamus et al., 2022; Daniluk et al., 2022; Błażkiewicz et al., 2021]. Therefore we decided to use it in this particular research.

  • BÅ‚ażkiewicz M, KÄ™dziorek J, Hadamus A. The Impact of Visual Input and Support Area Manipulation on Postural Control in Subjects after Osteoporotic Vertebral Fracture. Entropy. 2021; 23(3):375. https://doi.org/10.3390/e23030375
  • Casanova-Alvarez O, Morales-Helguera A, Cabrera-Pérez MÁ, Molina-Ruiz R, Molina C. A Novel Automated Framework for QSAR Modeling of Highly Imbalanced Leishmania High-Throughput Screening Data. J Chem Inf Model. 2021;61(7):3213-3231. doi: 10.1021/acs.jcim.0c01439
  • Daniluk A, Hadamus A, Ludwicki M, Zagrodny B. Backward vs. Forward Gait Symmetry Analysis Based on Plantar Pressure Mapping. Symmetry. 2022; 14(2):203. https://doi.org/10.3390/sym14020203
  • Dreher C, Kuder TA, König F, Mlynarska-Bujny A, Tenconi C, Paech D, Schlemmer HP, Ladd ME, Bickelhaupt S. Radiomics in diffusion data: a test-retest, inter- and intra-reader DWI phantom study. Clin Radiol. 2020;75(10):798.e13-798.e22. doi: 10.1016/j.crad.2020.06.024
  • Dunn OJ. Multiple comparisons using rank sums. Technometrics 1964;6:241–252.
  • Hadamus A, Jankowski T, Wiaderna K, Bugalska A, MarszaÅ‚ek W, BÅ‚ażkiewicz M, BiaÅ‚oszewski D. Effectiveness of Warm-Up Exercises with Tissue Flossing in Increasing Muscle Strength. Journal of Clinical Medicine. 2022; 11(20):6054. https://doi.org/10.3390/jcm11206054
  • Xu J, Cheng YJ, Wang ST, Wang X, Jin ZY, Qian TY, Zhu JX, Nickel MD, Xue HD. Simultaneous multi-slice accelerated diffusion-weighted imaging with higher spatial resolution for patients with liver metastases from neuroendocrine tumours. Clin Radiol. 2021;76(1):81.e11-81.e19. doi: 10.1016/j.crad.2020.08.024

We hope this explanation meets your requirements and that the manuscript will be accepted for publication in its current form.

Reviewer 4 Report

The authors sufficiently addressed all of my comments and improved their manuscript, especially in the method part. Therefore I recommend this article be published. In general, the article is of good EMG standard, including relevant sample size and methods.

Author Response

The authors sufficiently addressed all of my comments and improved their manuscript, especially in the method part. Therefore I recommend this article be published. In general, the article is of good EMG standard, including relevant sample size and methods

Thank you very much for this comment.